# Combined evaluation of Geriatric nutritional risk index and Neutrophil to lymphocyte ratio for predicting all-cause and cardiovascular mortality in hemodialysis patients

Jun Wang[1], Li-juan Huang[1], Bei Li[2], Mei-chang Xu[1], Lei Yang[1], Xu Deng[1], Xin Li[3]*

1 Department of Nephrology, Nanjing Integrated Traditional Chinese and Western Medicine Hospital, Nanjing, 210014, Jiangsu Province, China, 2 Department of Nephrology, Nanjing Hospital of Chinese Medicine Affiliated to Nanjing University of Chinese Medicine, Nanjing, 210012, Jiangsu Province, China, 3 Department of Science & Education Division, Nanjing Integrated Traditional Chinese and Western Medicine Hospital, Nanjing, 210014, Jiangsu Province, China

☯ These authors contributed equally to this work.

* fsyy00058@njucm.edu.cn

## Abstract

### Objective

Malnutrition, accompanied by an inflammatory profile, is a risk factor for poor prognosis in hemodialysis patients. The purpose of this study was to investigate the predictive value of NLR combined with GNRI for all-cause and cardiovascular mortality in hemodialysis patients.

### Methods

A total of 240 maintenance hemodialysis (MHD) patients in hemodialysis centers were enrolled in this retrospective study. The influencing factors of all-cause death in hemodialysis patients were analyzed by COX regression. The cut-off values of GNRI and NLR for predicting mortality in enrolled MHD patients were 89.01 and 4, respectively. Based on these cut-off values, the patients were divided into four groups: G1: high GNRI ($\geq$ 89.01) + high NLR ($\geq$ 4) group; G2: high GNRI ($\geq$ 89.01) + low NLR (<4) group, G3: low GNRI (< 89.01) + high NLR ($\geq$4) group; G4: low GNRI (< 89.01) + low NLR (<4).

### Results

During the follow-up period (average: 58 months), the all-cause mortality was 20.83%(50/240) and the cardiovascular mortality was 12.08%(29/240). Both NLR and GNRI were independent risk factors for the prognosis of MHD patients (P<0.05). Survival analysis showed that patients with low GNRI had a lower survival rate than those with high GNRI, whereas patients with high NLR had a lower survival rate than those with low NLR. Kaplan-Meier curve for all-cause mortality revealed that compared to G1, G2, and G4, G3 had the lowest survival rate, while G2 had the highest survival rate among all groups (P < 0.05). Kaplan-Meier curve for cardiovascular mortality showed that G3 had lower survival than G1, G2, and G4 (P < 0.001).

**Data Availability Statement:** All relevant data are within the paper and its Supporting Information files.

**Funding:** This research was funded by the Nanjing Administration of Traditional Chinese Medicine, Project name: Nanjing Traditional Chinese Medicine Youth Talent Training Program,(NO.ZYQ20053). The funders had no role in study design, data collection and analysis, decision to publish, or preparation of the manuscript.

**Competing interests:** The authors have declared that no competing interests exist.

## Conclusions

Our study demonstrates that bothGNRI and NLR are associated with all-cause mortality and cardiovascular mortality in MHD patients. Combining these two factorsmay contribute to a prognostic evaluation for MHD patients.

## 1. Introduction

With the improvement of medical and living standards, the problem of aging population is getting worse, while the incidence of chronic kidney disease (CKD) is increasing significantly [1]. End-stage kidney disease (ESRD) is the ultimate outcome for patients with CKD. Maintenance hemodialysis (MHD) is currently an essential treatment for ESRD patients [2]. ESRD patients with underlying conditions are at a high risk of death following hemodialysis [3]. Exploring the risk factors related to the mortality of MHD patients is a long-term task.

Due to the specificity of the disease itself, and the treatment for MHD patients, the incidence of protein-energy malnutrition is significantly increased, which is closely related to the poor prognosis [4]. The Geriatric Nutritional Risk Index (GNRI), a simple and effective tool for assessing nutritional status, has been validated as a simplified nutritional screening tool for hemodialysis patients, which is more easily prepared than other nutrition assessment tools [5]. It has been reported to be useful for MHD death prediction [6]. Besides, it is an excellent prognostic indicator [7]. Previous studies have shown an increase in all-cause, cardiovascular, and infectious mortality in MHD patients with malnutrition [8,9]. Early identification and intervention of malnutrition positively impact the prognosis of MHD patients. It is recommended that patients' nutritional status should be evaluated every 1–3 months to allow timely nutritional interventions [10]. Subjective global assessment and malnutrition inflammation score are prevalent in nutrition assessment [11,12]. However, the outcomes are prone to being affected by subjective factors in both methods. The GNRI, used to assess nutrition objectively based on height, weight and albumin levels, is easier to apply clinically [5]. In patients with multiple sclerosis, malnutrition is closely associated with the immune inflammatory response [13]. Neutrophils/lymphocytes ratio (NLR) is a novel marker of inflammation derived from routine blood count test. Studies have shown that the blood NLR in ulcerative colitis (UC) patients is associated with active diseases. The NLR can be used as the activity parameter of UC [14]. Another study showed that NLR could be used as a marker of diabetes control [15]. Studies have shown that NLR can also predict the prognosis of cancer patients [16]. Our previous studies have revealed that NLR and serum albumin indices are risk factors of mortality in MHD patients. NLR has been reported as an independent risk factor of mortality in MHD patients [17]. NLR levels in CKD3-5 patients gradually increase as kidney function declines, serving as an independent predictor of cardiovascular events [18]. The first prospective study in China that included CKD1-4 showed an association between NLR and ESRD progression risk in CKD4 patients; while no association was found between NLR and all-cause mortality in CKD1-3 patients [19].A prospective cohort study in Japan showed a significant increase in NLR in patients with end-stage renal disease receiving renal replacement therapy [20]. Previous studies have demonstrated that elevated NLR can increase the risk of cardiovascular disease in hemodialysis patients [21], NLR is an effective predictor of CVD death and all-cause death in hemodialysis patients [22], Previous studies have indicated that an NLR of 5.0 may be the threshold for the risk of patients with acute coronary syndrome [23]. Results of a meta-analysis suggest that NLR is a predictor of hospitalization and long-term outcome in patients

with acute ST segment elevation myocardial infarction after percutaneous coronary intervention [24]. At present, the association between all-cause mortality in NLR and MHD patients is still being explored and requires further investigation.

GNRI and NLR are clinically relevant indicators of mortality in MHD patients and contribute to early diagnosis and intervention. No studies have used GNRI and NLR combinations to predict prognosis in MHD patients. The study was based on data from two hemodialysis centers, focusing on the impact of GNRI and NLR on all-cause mortality in MHD patients.This study observed the prognostic value of combined application of GNRI and NLR in MHD patients, providing clinical basis for early clinical identification of MHD patients with poor prognosis.

## 2. Materials and methods

The study was based on data from two hemodialysis centers.From January 2014 to December 2017, 240 patients with initial hemodialysis were registered in the Chinese National Renal Data System (CNRDS) by Nanjing Integrated Traditional Chinese and Western Medicine Hospital and Nanjing Hospital of Chinese Medicine. The relevant Information and living status of patients were collected, and the relevant clinical indicators folloeing initial hemodialysis were queried in HIS (Hospital Information System). The patients were aged 26–96 years old with an average age of 63.4 ± 13.7 years old. The study was approved by the Ethics Committee of Nanjing Integrated Traditional Chinese and Western Medicine Hospital and Nanjing Hospital of Chinese Medicine (Ethics Approval Number 2022043).

### 2.1 Eligibility criteria

Inclusion criteria: (1) Meeting the criteria for ESRD [2]; (2) More than six months on dialysis; (3) Age 26–96 years old (4) Regular hemodialysis treatment; (5) Included in traditional Chinese medicine physical assessment.

Exclusion criteria: (1) Incomplete clinical data; (2)I Intermittent hemodialysis; (4) Treated with hemodialysis for acute renal injury.(5) Hemodialysis (HD) treatment for acute renal failure; (6) Irregular MHD treatment due to economic or family reasons;

### 2.2 Methods

Clinical Data and survival status of 240 patients with initial hemodialysis were found in the Chinese National Renal Data System (CNRDS) for retrospective study. Data were collected including sex, age, dialysis duration, education years,dialysis adequacy,vascular access types, primary disease,serum albumin,hemoglobin,pre-dialysis blood urea nitrogen,pre-dialysis serum creatinine,uric acid,triglyceride,total cholesterol,high-density lipoprotein, low-density lipoprotein, blood potassium,blood calcium,blood phosphorus,NLR,PLR,GNRI,body mass index.GNRI = 1.489 × rho (albumin) + 41.7 × (actual body mass / ideal body mass). The actual body mass is based on the dialysis dry weight. If the actual body mass / ideal body mass > 1, it was regarded as 1 [5]. NLR and PLR were calculated based on the blood routine indexes of the included patients. NLR represents the platelet/lymphocyte ratio, andPLR represents the neutrophil/lymphocyte ratio [17].

### 2.3 Study endpoint and follow-up

All-cause of death referred to death from any cause indicated study endpoint [25].

Cardiocerebrovascular death covered heart failure, myocardial infarction, cerebral hemorrhage, cerebral infarction, and peripheral vascular disease [25].

Follow-ups were conducted by outpatient, telephone, and home visits. The follow-up would be available before patients met the end point or until December 31, 2021.

### 2.4 Grouping

Based on these cut-off values, the patients were divided into four groups: G1: high GNRI ($\geq$ 89.01) + high NLR ($\geq$ 4) group; G2: high GNRI ($\geq$ 89.01) + low NLR ($<$4) group, G3: low GNRI ($<$ 89.01) + high NLR ($\geq$4) group; G4: low GNRI ($<$ 89.01) + low NLR ($<$4).

### 2.5 Outcome measures

To compare the clinical data of death and survival MHD patients, COX regression was used to analyze the risk factors for mortality in different groups. Predictive value of GNRI combined with NLR in MHD patients was assessed. The correlation between GNRI and NLR was analyzed. The effects of different levels of GNRI and NLR on MHD survival time were observed in groups according to the GNRI and NLR cut-off values of the included patients. Patients were grouped according to GNRI and NLR cutoff values to compare all-cause mortality and cardiovascular event mortality among different groups.

### 2.6 Statistical processing

Statistical processing was performed using IBM SPSS 24.0 and GraphPad Prism 7.0 software. Continuous variables were analyzed by independent samples t-test and expressed as mean ± standard deviation. Count data was processed with Pearson's chi square test. Classified variables were analyzed in two groups using the Kruskal-Wallis test or one-way variance analysis. Cox regression analysis was conducted for the risk factors of death in MHD patients. Factors included in this study were dialysis duration, years of education, the spKt/V, hemoglobin, blood phosphorus, platelet-lymphocyte ratio(PLR), GNRI, and NLR. Serum albumin and body mass index were excluded because they were used to establish GNRI indicators. The cut-off values of GNRI and NLR for death prediction in MHD patients were obtained according to ROC diagnostic curve. Based on GNRI and NLR cutoffs, survival analysis was performed using the Kaplan-Meier method, and inter-group differences were compared using the log-rank test. Person test was used to analyze the correlation between two variables;$P < 0.05$ was considered statistically significant.

## 3 Results

### 3.1 Comparison of general data

The study involved 161 male participants, with a mean age of 63.4 ± 13.7 years and an average dialysis duration of 64.8 ± 26.6 months. Vascular access types comprised arteriovenous fistula (183;76.25%), central venous catheters (21;8.75%), and artificial arteriovenous grafts (36;15.00%), Comorbid conditions included hypertension (225, 93.75%), diabetes (45; 18.75%), and heart disease (30; 12.50%). See **Table 1**.

### 3.2 COX regression analysis for all-cause death in MHD patients

During follow-up, 50 patients experienced all-cause death, including death from cardiovascular events (29/50), infections (13/50), malignancies (5/50), and other causes (3/50). Univariate COX regression analysis was used to identify GNRI and NLR levels as risk factors for all-cause mortality. COX Multifactor Regression Analysis incorporating dialysis duration, years of education, hemoglobin, blood phosphorus, PLR, indicated that GNRI and NLR were all risk

**Table 1. Comparison of related influencing factors of death in patients with MHD.**

| Categories | Death group(n = 50) | Survival group (n = 190) | χ2/t/Z value | P |
|---|---|---|---|---|
| Male [n (%)] | 37(74.00) | 125(65.79) | 1.216 | 0.270 |
| Age (years) | 63.3±13.6 | 64.1±14.1 | 0.405 | 0.686 |
| Dialysis duration (months) | 69.5±24.3 | 46.9±27.5 | 5.303 | <0.001 |
| Education years | 6.90±2.22 | 9.22±2.52 | 4.573 | <0.001 |
| Dialysis adequacy | 1.62±0.25 | 1.56±0.29 | 1.427 | 0.155 |
| Primary diseases | - | - | - | - |
| Chronic glomerulonephritis | 22(44.00) | 88(46.32) | 0.438 | 0.932 |
| Diabetic nephropathy | 10(20.00) | 40(21.05) | | |
| Hypertensive nephropathy | 7(14.00) | 28(14.74) | | |
| Other | 11(22.00) | 34(17.89) | | |
| Vascular access types | - | - | - | - |
| Arteriovenous fistula [n (%)] | 33(66.00) | 150(78.95) | 3.912 | 0.141 |
| Central venous catheters [n (%)] | 7(14.00) | 14(7.37) | | |
| Arteriovenous grafts [n (%)] | 10(20.00) | 26(13.68) | | |
| SA(g/L) | 37.28±4.86 | 39.79±3.40 | 3.430 | 0.001 |
| Hemoglobin (g/L) | 98.20±21.54 | 113.34±14.78 | 4.689 | <0.001 |
| Pre-dialysis blood urea nitrogen (mmol/L) | 26.85±6.51 | 26.06±7.69 | 0.733 | 0.465 |
| Pre-dialysis serum creatinine (umol/L) | 627.44±242.90 | 613.78±262.05 | 0.338 | 0.735 |
| Uric acid (umol/L) | 457.28±65.75 | 448.69±76.09 | 0.794 | 0.429 |
| Triglyceride (mmol/L) | 1.96±0.46 | 1.91±0.55 | 0.552 | 0.582 |
| Total cholesterol (mmol/L) | 3.94±0.66 | 3.85±0.79 | 0.777 | 0.439 |
| High-density lipoprotein (mmol/L) | 1.08±0.31 | 1.04±0.37 | 0.623 | 0.534 |
| Low-density lipoprotein (mmol/L) | 3.46±0.64 | 3.36±0.75 | 0.762 | 0.448 |
| Blood potassium (mmol/L) | 4.65±0.69 | 4.55±0.79 | 0.809 | 0.421 |
| Blood calcium (mmol/L) | 2.07±0.20 | 2.05±0.23 | 0.696 | 0.488 |
| Blood phosphorus (mmol/L) | 1.55±0.58 | 1.97±0.43 | 4.743 | <0.001 |
| NLR | 3.69±1.58 | 2.94±1.09 | 3.154 | 0.003 |
| PLR | 125.97±17.18 | 117.19±13.94 | 3.336 | 0.001 |
| GNRI | 85.82±10.75 | 93.53±6.67 | 4.837 | <0.001 |
| Body mass index (kg/m2) | 18.89±3.34 | 21.15±2.38 | 4.497 | <0.001 |

Note: SA, serum albumin; NLR, neutrophil/lymphocyte ratio; PLR, platelet/lymphocyte ratio; GNRI: Geriatric nutritional risk index; MHD, maintenance hemodialysis.

factors for MHD-related death. Serum albumin and body mass index were excluded because they were used to establish GNRI indicators (**Table 2**).

## 3.3 Predictive value of GNRI and NLR for mortality in MHD patients

The area under the ROC curve for GNRI's prediction for mortality in MHD patients was 0.742. At the GNRI threshold of 89.01, its Jordon index was 0.506, with a specificity of 0.680

**Table 2. COX Multifactor analysis of prognosis in MHD patients.**

| Variables | β | SE | Wald value | OR value (95%CI) | P |
|---|---|---|---|---|---|
| NLR | 0.563 | 0.139 | 16.532 | 1.757(1.339–2.305) | <0.001 |
| GNRI | -0.138 | 0.027 | 26.342 | 0.871(0.827–0.918) | <0.001 |

Note: NLR, neutrophil/lymphocyte ratio; GNRI: Geriatric nutritional risk index.

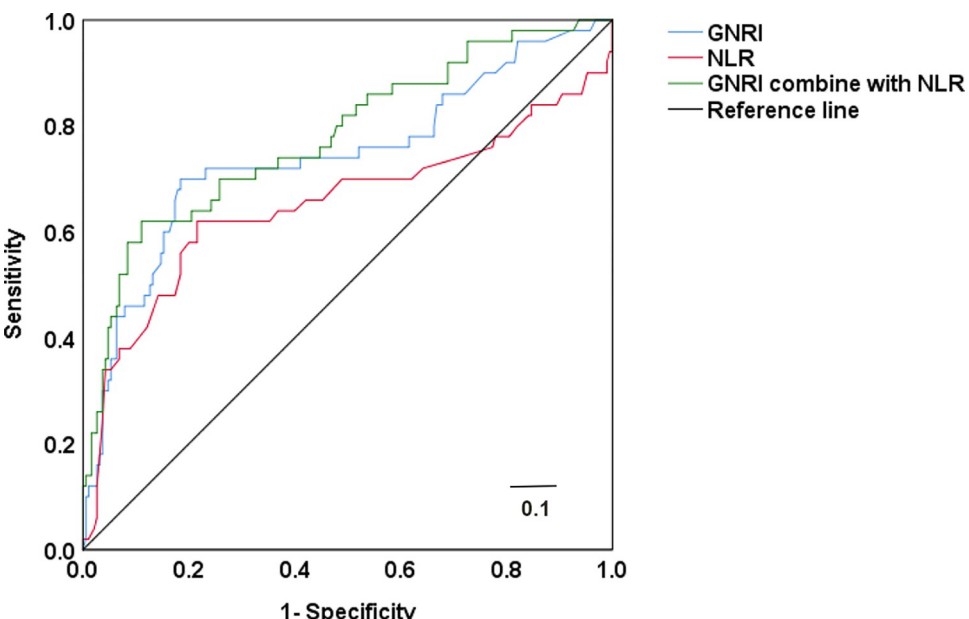

**Fig 1. Predictive value of GNRI and NLR for mortality in MHD patients.** Abbreviations: GNRI: Geriatric nutritional risk index, NLR: Neutrophils / lymphocytes ratio.

and a sensitivity of 0.826. The area under the ROC curve for the combined GNRI and NLR's prediction for mortality in MHD patients was 0.652. At the NLR threshold of 4.0, its Jordon index was 0.404, with a specificity of 0.784 and a sensitivity of 0.620. The equation of GNRI combined with NLR = $\text{Log}^{(7.623 = 0.307*NLR-0.111*GNRI)}$. The area under the ROC curve for the combined GNRI and NLR's prediction for mortality in MHD patients was 0.780, with a Jordon index of 0.509, a specificity of 0.620 and a sensitivity of 0.889. See Fig 1.

### 3.4 GNRI correlates with NLR

GNRI was negatively correlated with NLR (r = -0.229), as shown in Fig 2.

### 3.5 All-cause death survival curves of MHD patients with different levels of GNRI, NLR

The diagnostic threshold of GNRI at 89.01 was regarded as a cutoff value. A total of 175 patients with high GNRI (mortality19/175) and 65 patients with low GNRI (mortality 31/ 65) were classified as statistically significant differences. Survival analysis revealed that the survival rate of patients with low GNRI was lower than those with high GNRI (($\chi^2$ = 48.712, P<0.001)) (**Fig 3A**). The NLR diagnostic threshold of 4.0 was considered as a dividing line. The patients were categorized into high NLR group with 69 patients (mortality 31/ 69) and low NLR group with 171 patients (mortality 19/171). Survival analysis indicated that patients with high NLR survival has lower survival rate than those with low NLR ($\chi^2$ = 48.712, P < 0.001) (**Fig 3B**).

### 3.6 Survival curve analysis of different subgroups and all causes of death and cardiovascular events

In the all-cause mortality comparison, there were 45 cases in G1 (mortality 8 / 45), 128 cases in G2 (mortality 10 / 128), 27 cases in G3 (mortality 23 / 27) and 40 cases in G4 (mortality 9 / 40) (Table 3). Survival analysis revealed that G3 group had a lower survival rate than G1, G2, and

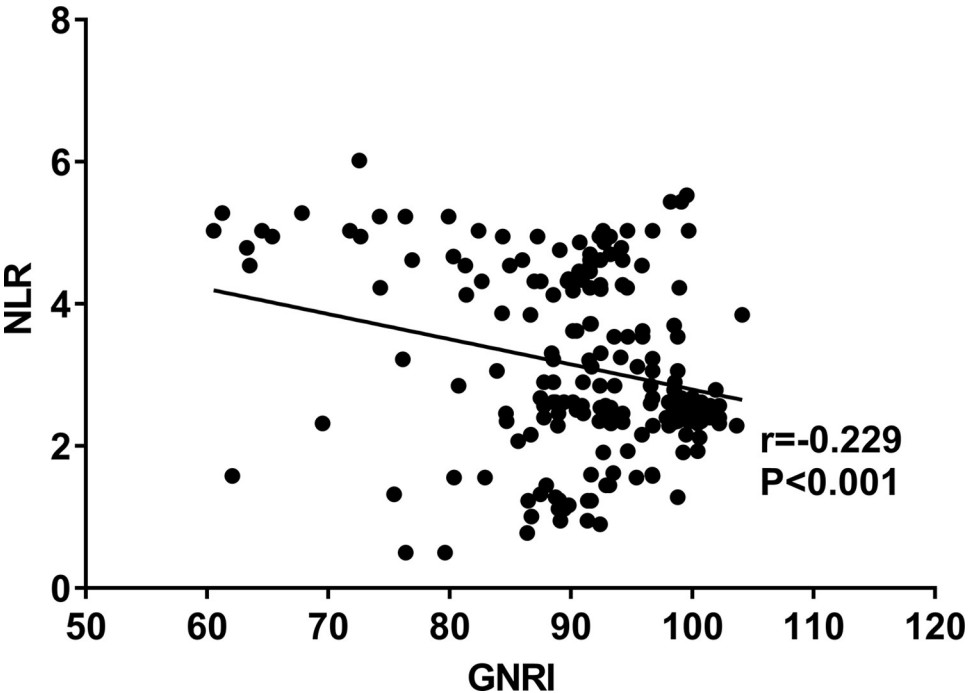

**Fig 2. GNRI correlates with NLR.** Abbreviations: GNRI: Geriatric nutritional risk index, NLR: Neutrophils / lymphocytes ratio.

G4 groups (P < 0.001), while G2 group exhibited a higher survival than G1, G3 and G4 (P < 0.05) (**Fig 4A**). Cardiovascular events accounted for 29 of the 50 deaths. There were 45 cases in G1 (mortality 4/ 45), 128 cases in G2 (mortality 6 / 128), 27 cases in G3 (mortality 14/ 27) and 40 cases in G4 (mortality 5/ 40) (**Table 3**). Survival analysis demonstrated a lower survival rate in G3 compared to G1, G2, and G4 (P < 0.001) (**Fig 4B**).

## 4 Discussion

There were statistical differences in the indicators (dialysis duration, education years, hemoglobin, blood phosphorus, NLR, PLR and GNRI) among different groups. It has been reported

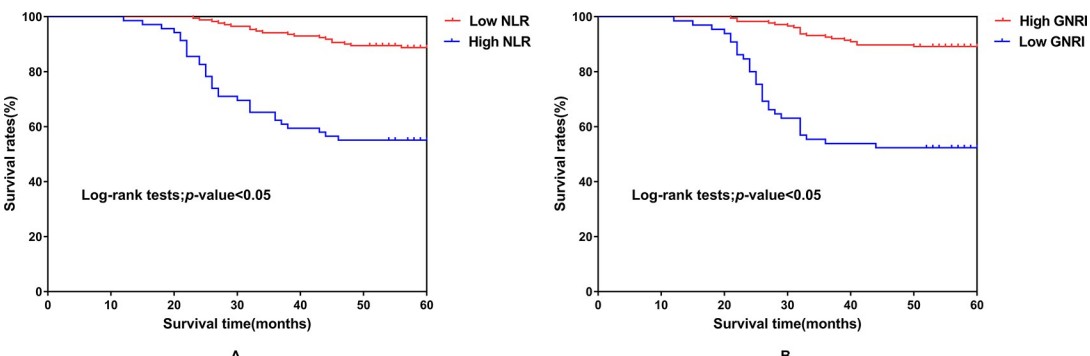

**Fig 3. Kaplan–Meier survival curves for all-cause mortality.** Comparison of different GNRI levels with all-cause mortality (A). Comparison of different NLR levels with all-cause mortality (B). Abbreviations: Ref: Reference values, GNRI: Geriatric nutritional risk index, NLR: Neutrophils / lymphocytes ratio.

**Table 3. A comparison of mortality rates in different groups.**

| Categories | G1 (n = 45) | G2 (n = 128) | G3 (n = 27) | G4 (n = 40) | χ2 value | P |
|---|---|---|---|---|---|---|
| All-cause death [n (%)] | 8(17.78) | 10(7.81) | 23(85.19) ***###@@@ | 9(22.50)&&& | 81.273 | <0.001 |
| Cardiovascular mortality [n (%)] | 4(8.89) | 6 (4.69) | 14(60.87)***###@@@ | 5(12.50)&&& | | |
| Non-cardiovascular mortality [n (%)] | 4(8.89) | 4(3.13) | 9(33.33)***@@@ | 4(10.00)&&& | 81.800 | <0.001 |

Note:*Compared with G1and G3,***P<0.001;#Compared with G2 and G3,### P<0.001;@Compared with G3and G4,@@@ P<0.001;&Compared with G2and G4, &&& 为P<0.001.

that dialysis duration is positively correlated with the occurrence of sarcopenia in MHD patients [26]. The sarcopenia leads to a decrease in exercise capacity, resulting in an increased incidence of cardiovascular and cerebrovascular events. With the increase of dialysis duration, the patients experience more nutrient loss, acid-base imbalance, and hormonal level changes, resulting in an increased incidence of sarcopenia [27]. Patients with lower years of education are more likely to be frail due to lower incomes, which may indirectly affect their ability to receive better treatment [28]. The dialysis adequacy refers to the removal of excess water and toxins from the patient's body through hemodialysis to achieve a comfortable state [29]. Indicators such as serum albumin, hemoglobin, and body mass index reflect the nutritional status of patients. The risk of frailty is 1.89 times higher in hemodialysis (MHD) patients with serum albumin concentrations below 32 g/L than in those with concentrations at or above 39 g/L [30]. Lower hemoglobin concentration in MHD patients,are associated with a higher likelihood of debilitation and poor prognosis [31]. MHD patients with a low body mass index ($<18.5$ kg/m$^2$) have a significantly increased risk of frailty, sarcopenia, and consequently, a higher risk of death [32,33]. Grip strength, grip strength, walking speed, and upper arm circumference serve as indicators of muscle strength and mass. Studies have shown that the prevalence of sarcopenia in MHD patients ranges between 3.9% to 63.3% [34,35]. Furthermore, sarcopenia can induce an increased incidence of adverse events and affect prognosis [36,37].

In this study, patients were grouped based on the cut-off values of GNRI and NLR for MHD death prediction (89.01 and 4, respectively). The 5-year follow-up of 240 MHD patients included in this study showed 50 all-cause deaths, with a 5-year mortality rate of 20.83%. By multifactorial Cox regression analysis, NLR and GNRI were found as independent risk factors for prognosis in MHD patients. NLR is a systemic inflammation marker. Inflammatory factors

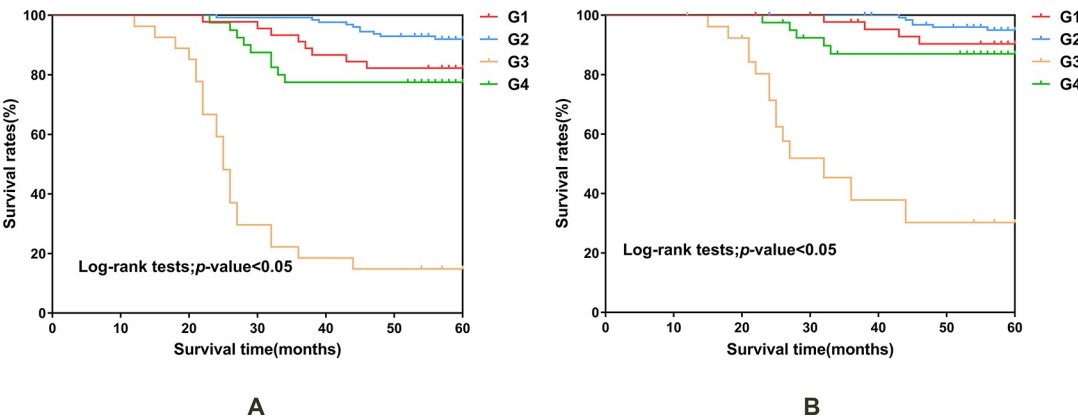

**Fig 4. Survival curve analysis of different groups and all-cause death and cardiovascular mortality.** Comparison of mortality rates from cause among groups (A). Comparison of mortality rates from cardiovascular events among groups (B).

easily trigger malnutrition and thereby induce a debilitating syndrome [38]. In our study, NLR and PLR levels were increased in MHD patients with the debilitating syndrome, and NLR was an independent risk factor for prognosis in MHD patients [17]. As an important indicator of systemic inflammation, NLR is an independent risk factor for predicting kidney failure in patients with stage 4 chronic kidney disease [19]. Stimulated by inflammation, it induces hyperplasia of megakaryocytes, which in turn interacts with endothelial cells and white blood cells to produce inflammatory factors in a vicious cycle. The occurrence of various diseases is associated with increased platelets [39,40]. NLR is a novel inflammatory factor that plays a great role in the development of coronary heart disease, myocardial infarction and tumors [41,42]. Mortality in MHD patients may be linked to higher NLR levels promoting tumor and cardiovascular disease.

GNRI is one of the indicators of nutritional assessment. Its use in Asian MHD patients has some predictive value for nutritional status and prognosis [42,43]. A study involving 3,536 MHD patients points out that low levels of GNRI are associated with increased mortality from all causes [44]. Another study of 2,791 people with chronic kidney disease indicates that low GNRI levels are connected to cardiovascular and all-cause mortality [7]. Poor immune function due to malnourishment increases the incidence of inflammatory infections, leading to cardiovascular events and poor prognosis. In this study, the relationship between GNRI and NLR reveals a negative correlation between them, which may be related to nutrient loss in patients with inflammatory factors.

In this study, GNRI combined with NLR markers significantly increased the predictive value of death. We subdivided MHD patients with varying degrees of GNRI and NLR, with 89.01 of GNRI and 4.0 of NLR as cutoff values for death prediction, respectively. This indicated lower survival in patients with high NLR and low GNRI. Kobayashi's study shows chronic inflammatory levels in MHD patients with GNRI$\geq$90 have lower all-cause mortality compared to those with GNRI<90 [45]. Another clinical study reveals a significant increase in all-cause mortality in MHD patients with GNRI below 92 [46]. Epidemiologically, GNRI predicts cardiovascular outcomes, and studies have shown a correlation between GNRI and cardiovascular events in patients with chronic heart failure [47]. It has been verified that GNRI is strongly associated with mortality from various causes and cardiovascular events in MHD patients [7]. Chronic inflammation is common in MHD patients. Inflammatory states increase muscle protein metabolism, inhibit albumin production and accelerate its breakdown [48]. Our previous study showed an increase in frailty and mortality in MHD patients with NLR above 2.98 [17]. Another study showed higher NLR levels result in more cardiovascular events in MHD patients [49].

Based on these findings, we further subdivided the GNRI and NLR levels The low GNRI + high NLR group (G3) had the worst all-cause mortality and cardiovascular mortality, while the high GNRI + low NLR group(G2) had the best all-cause mortality compared to the other three groups. Studies have revealed a negative correlation between GNRI and leptin, which predicts patient's future nutritional status [50]. MHD patients with high GNRI indicate good nutritional status and higher lean mass index and bone mineral density [51]. It has been reported that there is a negative correlation between GNRI levels and IL-6 in MHD patients [52]. Elevated NLR levels in patients indicate an increase in neutrophils and a decrease in lymphocytes, which are pro-inflammatory cells releasing pro-inflammatory factors, activating macrophages to promote foam cell formation, and inducing cardiovascular diseases [53]. Decreased lymphocytes and a weakened immune system can trigger infection and cardiovascular events [54]. The combination of GNRI and creatinine index was used to predict all-cause mortality of MHD. Lower GNRI and Cr indices were associated with an increased risk of all-cause mortality, and GNRI was considered more suitable for predicting prognosis in

hemodialysis patients because it is simpler to calculate than creatinine index [44]. In addition, MHD patients with low GNRI and low modified creatinine index had significantly higher all-cause mortality and cardiovascular event mortality [55]. Consequently, we suggest that when MHD patients are in a low GNRI and high NLR state, the body's immunity weakens, and infections and cardiovascular events increase, leading to poor prognosis.

However, there are some limitations in this study. (1) GNRI and NLR may change over time. (2) Other influencing factors weren't considered in this retrospective study. (3) The sample size was small. Further expansion of the sample should be carried out to determine the association between GNRI and NLR for all-cause and cardiovascular mortality.

In conclusion, our study shows that GNRI and NLR are associated with all-cause mortality and cardiovascular mortality in MHD patients. A combination of the two may contribute to a prognostic evaluation in MHD patients.

## Supporting information

**S1 File. Original data.**
(XLS)

## Author Contributions

**Data curation:** Bei Li, Xu Deng, Xin Li.

**Formal analysis:** Jun Wang, Bei Li.

**Methodology:** Li-juan Huang.

**Validation:** Li-juan Huang, Mei-chang Xu, Lei Yang.

**Writing – original draft:** Xin Li.

**Writing – review & editing:** Jun Wang.

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
