## [Decision Letter · Decision Letter 0]

23 Mar 2023

PONE-D-23-04816Combined Evaluation of Geriatric Nutritional Risk Index and Neutrophil to lymphocyte ratio for Predicting All-cause mortality and cardiovascular mortality in Hemodialysis PatientsPLOS ONE

Dear Dr. Wang,

Thank you for submitting your manuscript to PLOS ONE. After careful consideration, we feel that it has merit but does not fully meet PLOS ONE’s publication criteria as it currently stands. Therefore, we invite you to submit a revised version of the manuscript that addresses the points raised during the review process.

We look forward to receiving your revised manuscript.

Kind regards,

Gulali Aktas

Academic Editor

PLOS ONE

Journal Requirements:

   "Funding: This research was funded by the Nanjing Administration of Traditional Chinese Medicine, Project name: Nanjing Traditional Chinese Medicine     Youth Talent Training Program,(NO.ZYQ20053).

    YES - Specify the role(s) played."

Additional Editor Comments:

Make revisions in accordance with the reviewers, please.

I invite to respond the comments of the reviewers. the paper will be reconsidered if adequately revised.

Reviewers' comments:

Reviewer's Responses to Questions

**Comments to the Author**

1. Is the manuscript technically sound, and do the data support the conclusions?

Reviewer #1: Yes

Reviewer #2: Partly

2. Has the statistical analysis been performed appropriately and rigorously? 

Reviewer #1: Yes

Reviewer #2: I Don't Know

3. Have the authors made all data underlying the findings in their manuscript fully available?

Reviewer #1: Yes

Reviewer #2: Yes

4. Is the manuscript presented in an intelligible fashion and written in standard English?

Reviewer #1: Yes

Reviewer #2: No

5. Review Comments to the Author

Reviewer #1: Introduction; I advise authors to make it clear why they studied NLR in hemodialysis patients as a risk factor for CV mortality. Chronic kidney disease is associated with low grade inflammation (). On the other hand, NLR is associated with various inflammatory conditions such as pancreatitis (Iran J Colorectal Res 2022;10(3):75-86.), diabetes mellitus type 2 (African health sciences, 19(1), 1602-1606), autoimmune disorders (Revista da Associação Médica Brasileira, 63, 1065-1068), and thyroid conditions (Ethiop. J. Health Dev. 2021; 35(3):149-153). Inflammation involve in all of these diseases similar to the chronic kidney disease.

Materials&Methods; An inclusion criteria was noted as Age≥18. However, mean age was 63 years. Please revise.

Results and Discussion; fair enough

Reviewer #2: 1. the title is not well worded and too long ("All-cause mortality and cardiovascular mortality"?????)

2. Abstract: in Introduction add: "Aim of the study was To investigate the predictive value of the.........."

3. Abstract: methodology - this sentence do not belong here: "COX regression analysis showed that GNRI and NLR were risk factors for death in MHD patient" same one is in the Results section too.

4. Improve English

6. PLOS authors have the option to publish the peer review history of their article (what does this mean?). If published, this will include your full peer review and any attached files.

Reviewer #1: No

Reviewer #2: No

---

## [Author Response · Author response to Decision Letter 0]

24 Apr 2023

Response the academic editor：

Response：We have revised the manuscript in accordance with the article format required by POLS ONE.

2.Thank you for stating the following financial disclosure:

   "Funding: This research was funded by the Nanjing Administration of Traditional Chinese Medicine, Project name: Nanjing Traditional Chinese Medicine     Youth Talent Training Program,(NO.ZYQ20053).

    YES - Specify the role(s) played."

Response：Annotations have been added to the manuscript

Response：Article data can be used publicly, the original data has been provided to the editorial department when submission, we have changed in the declaration.

Response：We have deleted the relevant content from the article.

Response：We have supplemented the original data used in the article and have no other supplementary information.

Response Reviewer #1：

①Introduction; I advise authors to make it clear why they studied NLR in hemodialysis patients as a risk factor for CV mortality. Chronic kidney disease is associated with low grade inflammation (). On the other hand, NLR is associated with various inflammatory conditions such as pancreatitis (Iran J Colorectal Res 2022;10(3):75-86.), diabetes mellitus type 2 (African health sciences, 19(1), 1602-1606), autoimmune disorders (Revista da Associação Médica Brasileira, 63, 1065-1068), and thyroid conditions (Ethiop. J. Health Dev. 2021; 35(3):149-153). Inflammation involve in all of these diseases similar to the chronic kidney disease.

Response：Thank you for your suggestion.We have added relevant content in the manuscript.

Materials&Methods; An inclusion criteria was noted as Age≥18. However, mean age was 63 years. Please revise.

Response：Thank you for your suggestion.We have revised it in the manuscript.

Response Reviewer #2：

1.the title is not well worded and too long ("All-cause mortality and cardiovascular mortality"?????)

Response：Thank you for your suggestion.We have added relevant content in the manuscript.

2. Abstract: in Introduction add: "Aim of the study was To investigate the predictive value of the.........."

Response：Thank you for your suggestion.We have revised it in the manuscript.

3. Abstract: methodology - this sentence do not belong here: "COX regression analysis showed that GNRI and NLR were risk factors for death in MHD patient" same one is in the Results section too.

Response：Thank you for your suggestion.We have revised it in the manuscript.

4. Improve English

Response：Thank you for your suggestion.We have revised it in the manuscript.

---

## [Decision Letter · Decision Letter 1]

1 Jun 2023

PONE-D-23-04816R1Combined Evaluation of Geriatric Nutritional Risk Index and Neutrophil to lymphocyte ratio for Predicting All-cause mortality and cardiovascular mortality in Hemodialysis PatientsPLOS ONE

Dear Dr. Wang,

Thank you for submitting your manuscript to PLOS ONE. After careful consideration, we feel that it has merit but does not fully meet PLOS ONE’s publication criteria as it currently stands. Therefore, we invite you to submit a revised version of the manuscript that addresses the points raised during the review process.

We look forward to receiving your revised manuscript.

Kind regards,

Gulali Aktas

Academic Editor

PLOS ONE

Journal Requirements:

Reviewers' comments:

Reviewer's Responses to Questions

**Comments to the Author**

1. If the authors have adequately addressed your comments raised in a previous round of review and you feel that this manuscript is now acceptable for publication, you may indicate that here to bypass the “Comments to the Author” section, enter your conflict of interest statement in the “Confidential to Editor” section, and submit your "Accept" recommendation.

Reviewer #1: All comments have been addressed

Reviewer #2: All comments have been addressed

2. Is the manuscript technically sound, and do the data support the conclusions?

Reviewer #1: Yes

Reviewer #2: Yes

3. Has the statistical analysis been performed appropriately and rigorously? 

Reviewer #1: Yes

Reviewer #2: I Don't Know

4. Have the authors made all data underlying the findings in their manuscript fully available?

Reviewer #1: Yes

Reviewer #2: Yes

5. Is the manuscript presented in an intelligible fashion and written in standard English?

Reviewer #1: Yes

Reviewer #2: Yes

6. Review Comments to the Author

Reviewer #1: Dear Author

You have mentioned a very good topic in your article but background must be improved. NLR is a novel marker of inflammation derived from routine blood count test. Its association with inflammation has been reported in inflammatory bowel disease (Wiener klinische Wochenschrift (2015)127:262-265. DOI 10.1007/s00508-014-0683-5), diabetes mellitus (Afri Health Sci. 2019;19(1):1602-1606. DOI: 10.4314/ahs.v19i1.35), and gastrointestinal conditions (Iran J Colorectal Res. 2022;10(3):75-86. doi: 10.30476/ACRR.2022.97244.1160). Improve the background please.

Best Regards...

Reviewer #2: - In Abstract section, please write full terms for NLR and GNRI.

- In Results section, for number of patients (50 and 29) except total number write values in percentages (%)

- Three last sentences in Introduction should be written on better way and they should have better formulation in context of Introduction ("No studies have used GNRI and NLR combinations to predict prognosis in MHD patients. GNRI and NLR are clinically relevant indicators of mortality in MHD patients and contribute to early diagnosis and intervention. The study was based on data from two hemodialysis centers, focusing on the impact of GNRI and NLR on all-cause mortality in MHD patients."

- "The study was based on data from two hemodialysis centers" - this information should be part of Materials and Methods

- In Materials and Methods section write the type of the study

- Inclusion and Exclusion criteria should include more information about patients

- What about Inform consent of the patients? Did you provide it?

- In Methods section include formula for NLR calculation

- In Materials and Methods section include information about all parameters that you was analyzed in the study and explain (for example: hemoglobin, blood phosphorus, platelet-lymphocyte ratio(PLR)...)

- In Statistical processing section, explain how you determined correlation between two variables (Did you use Pearson or Sperman correlation coefficient?)

7. PLOS authors have the option to publish the peer review history of their article (what does this mean?). If published, this will include your full peer review and any attached files.

Reviewer #1: No

Reviewer #2: No

---

## [Author Response · Author response to Decision Letter 1]

7 Jun 2023

Response Reviewer #1：

1.You have mentioned a very good topic in your article but background must be improved. NLR is a novel marker of inflammation derived from routine blood count test. Its association with inflammation has been reported in inflammatory bowel disease (Wiener klinische Wochenschrift (2015)127:262-265. DOI 10.1007/s00508-014-0683-5), diabetes mellitus (Afri Health Sci. 2019;19(1):1602-1606. DOI: 10.4314/ahs.v19i1.35), and gastrointestinal conditions (Iran J Colorectal Res. 2022;10(3):75-86. doi: 10.30476/ACRR.2022.97244.1160). Improve the background please.

Best Regards...

Response：Thank you for your suggestion.We have added relevant content in the manuscript.

Response Reviewer #2： 

1.In Results section, for number of patients (50 and 29) except total number write values in percentages (%)

Response：Thank you for your suggestion.We have revised it in the manuscript.

2. Three last sentences in Introduction should be written on better way and they should have better formulation in context of Introduction ("No studies have used GNRI and NLR combinations to predict prognosis in MHD patients. GNRI and NLR are clinically relevant indicators of mortality in MHD patients and contribute to early diagnosis and intervention. The study was based on data from two hemodialysis centers, focusing on the impact of GNRI and NLR on all-cause mortality in MHD patients."

Response：Thank you for your suggestion.We have revised it in the manuscript.

3.  "The study was based on data from two hemodialysis centers" - this information should be part of Materials and Methods

Response：Thank you for your suggestion.We have revised it in the manuscript.

4. In Materials and Methods section write the type of the study

Response：Thank you for your suggestion.We have revised it in the manuscript.

5.Inclusion and Exclusion criteria should include more information about patients

Response：Thank you for your suggestion.We have revised it in the manuscript.

6.What about Inform consent of the patients? Did you provide it?

Response：Due to the retrospective nature of the study, informed conssent was waived.The study was approved by the ethic committee of our hospital.

7.In Methods section include formula for NLR calculation

Response：Thank you for your suggestion.We have revised it in the manuscript.

8.In Materials and Methods section include information about all parameters that you was analyzed in the study and explain (for example: hemoglobin, blood phosphorus, platelet-lymphocyte ratio(PLR)...)

Response：Thank you for your suggestion.We have revised it in the manuscript.

9.In Statistical processing section, explain how you determined correlation between two variables (Did you use Pearson or Sperman correlation coefficient?)

Response：Person test was used to analyze the correlation between two variables.Modifications were made in statistical methods.

---

## [Editor Report · Decision Letter 2]

12 Jun 2023

Combined Evaluation of Geriatric Nutritional Risk Index and Neutrophil to lymphocyte ratio for predicting all-cause and cardiovascular mortality in hemodialysis patients

PONE-D-23-04816R2

Dear Dr. Li,

We’re pleased to inform you that your manuscript has been judged scientifically suitable for publication and will be formally accepted for publication once it meets all outstanding technical requirements.

Kind regards,

Gulali Aktas

Academic Editor

PLOS ONE

Additional Editor Comments (optional):

Dear Authors

You responded the editorial and reviewers' comments adequately. There is nothing more that need further revision. Thank you.
---

## [Editor Report · Acceptance letter]

22 Jun 2023

PONE-D-23-04816R2 

Combined Evaluation of Geriatric Nutritional Risk Index and Neutrophil to lymphocyte ratio for predicting all-cause and cardiovascular mortality in hemodialysis patients 

Dear Dr. Li:

I'm pleased to inform you that your manuscript has been deemed suitable for publication in PLOS ONE. Congratulations! Your manuscript is now with our production department. 

Kind regards, 

on behalf of

Professor Gulali Aktas 

Academic Editor

PLOS ONE